# Cosmic ray interactions in the atmosphere: QGSJET-III and other models

**Sergey Ostapchenko**

Universität Hamburg, II Institut für Theoretische Physik, 22761 Hamburg, Germany
D.V. Skobeltsyn Institute of Nuclear Physics, Moscow State University,
119992 Moscow, Russia

sergey.ostapchenko@desy.de

*21st International Symposium on Very High Energy Cosmic Ray Interactions
(ISVHECRI 2022)
Online, 23-28 May 2022*

## Abstract

**The physics content of the QGSJET-III Monte Carlo generator of high energy hadronic collisions is discussed. New theoretical approaches implemented in QGSJET-III are addressed in some detail and a comparison to alternative treatments of other cosmic ray interaction models is performed. Calculated characteristics of cosmic ray-induced extensive air showers are presented and differences between the respective results of QGSJET-III and other models are analyzed. In particular, it is demonstrated that those differences are partly caused by severe deficiencies of the other interaction models.**

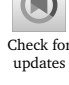
## 1 Introduction

Studies of cosmic rays (CRs) of very high energies are traditionally performed using the extensive air shower (EAS) techniques: measuring characteristics of nuclear-electromagnetic cascades induced by interactions of the primary CRs in the atmosphere. Consequently, a necessary ingredient of the corresponding experimental analysis procedures are numerical simulations of EAS development. A special role in such simulations is played by Monte Carlo (MC) generators of high energy hadronic interactions, designed to treat inelastic collisions with air nuclei of both the primary CR particles and of secondary hadrons produced in the course of EAS development.

Since general collisions of hadrons and nuclei can not be fully described within the perturbative framework, such MC generators necessarily involve phenomenological approaches. Therefore, the role of the calibration of CR interaction models, based on available accelerator data, notably, from the Large Hadron Collider (LHC), is difficult to overestimate. On the

other hand, of equal importance is an overall self-consistency of the underlying theoretical approaches employed in such models. Needless to say, respecting the relevant conservation laws, e.g. regarding energy-momentum and electric charge, and symmetries (e.g. isospin[1]) is a must. Doing otherwise leads one to incorrect predictions for various relevant characteristics of hadronic interactions and may introduce an important bias into CR data analyses.

## 2 Quark-gluon string and dual parton models

All present MC generators of hadronic interactions are based on the qualitative picture of quantum chromodynamics (QCD): the collisions between both hadrons and nuclei are mediated by cascades of partons [(anti)quarks and gluons]. It is important to keep in mind that interacting hadrons form their parton "coats" well before the collision: by emitting multiple virtual parton cascades. When some partons from the projectile "cloud" meet their counterparts from the target and scatter of each other, the scattering may destroy the coherence of the initial parton fluctuation, causing inelastic rescattering processes giving rise to secondary hadron production. Alternatively, the coherence of some virtual parton cascades may be preserved by the scattering process and the corresponding partons will recombine back to their parent hadrons, which corresponds to elastic rescattering processes. Thus, a hadron-hadron interaction generally contains multiple inelastic and elastic rescattering processes.

It is customary to use the eikonal approximation for treating multiple scattering: assuming all the inelastic and elastic processes to be independent of each other, for a given impact parameter $b$ between the interacting hadrons. This leads one to simple expressions for the interaction cross sections, e.g. for the total proton-proton cross section one obtains

$$\sigma_{pp}^{\text{tot}}(s) = 2 \int d^2 b \left[ 1 - e^{-\chi_{pp}(s,b)} \right], \tag{1}$$

where $s$ is the center-of-mass (c.m.) energy squared for the collision and the eikonal $\chi_{pp}(s, b)$ is defined by the imaginary part of the scattering amplitude for a single scattering process.

A very successful description of high energy hadronic collisions had been provided by the Quark-Gluon String [1] and Dual Parton [2] models developed within the Reggeon Field Theory (RFT) framework [3]. An elementary rescattering process has been described by a Pomeron exchange, with the respective eikonal $\chi_{pp}^{\mathbb{P}}$ having only 3 adjustable parameters:

$$\chi_{pp}^{\mathbb{P}}(s, b) = \frac{\gamma_p^2 s^\Delta}{2R_p^2 + \alpha_{\mathbb{P}}' \ln s} \exp\left[ -\frac{b^2/4}{2R_p^2 + \alpha_{\mathbb{P}}' \ln s} \right]. \tag{2}$$

The so-called overcriticality $\Delta > 0$ controls the energy-rise of $\chi_{pp}^{\mathbb{P}}$, reflecting the energy-rise of the parton density, while the Pomeron slope $\alpha_{\mathbb{P}}'$ is related to parton transverse diffusion.

The conversion of partons into secondary hadrons involved the concept of the color exchange: after the collision, constituent partons [(anti)quarks and (anti)diquarks] from the interacting hadrons appeared to be connected to each other by tubes (strings) of color field. With the partons flying apart, the tension of the string rises until it breaks, with the color field being neutralized via a creation of additional quark-antiquark and diquark-antidiquark pairs from the vacuum, giving rise to a formation of secondary hadrons.

---

[1]While the isospin symmetry is not an exact one for strong interactions, it holds to a very good accuracy thanks to the small mass difference between the $u$ and $d$ quarks.

# 3 From QGSJET to QGSJET-III

## 3.1 Combined treatment of soft and hard processes

The traditional RFT assumes hadronic collisions to be dominated by pure soft processes, corresponding to production of hadrons of relatively low transverse momenta $p_\perp \lesssim 1$ GeV. On the other hand, with increasing energy, the so-called semihard processes involving cascades of high $p_\perp$ partons become more and more important [4]. This is because the smallness of the respective strong coupling $\alpha_s(p_\perp^2)$ becomes compensated by large collinear and infrared logarithms: the logarithmic ratios of transverse $\ln(p_{\perp_i}^2/p_{\perp_{i-1}}^2)$ and longitudinal $\ln(x_{i-1}/x_i)$ momenta of subsequent partons [$x_i$ being the fraction of the parent hadron light cone (LC) momentum, carried by $i$-th parton] in the corresponding parton cascades preceding the hardest (highest $p_\perp$) parton-parton scattering.

The QGSJET model [5,6] was designed to treat both soft and semihard processes coherently within the RFT framework, based on the "semihard Pomeron" approach [5,7,8]. The main idea was to employ the perturbative QCD (pQCD) formalism for treating perturbative parts of the underlying parton cascades, for parton virtualities $|q^2|$ above some chosen cutoff $Q_0^2$ for pQCD being applicable, while keeping the Pomeron description for pure soft ($|q^2| < Q_0^2$) processes and for soft parts of semihard parton cascades. This allowed one to develop the Pomeron calculus, based on the "general Pomeron" which thus combines the soft and semihard contributions.

With regard to EAS modeling, the main consequence of taking semihard processes into account was a steeper energy rise of the multiple scattering rate: $\propto s^{\Delta_{\text{hard}}}$, $\Delta_{\text{hard}} \simeq 0.3$, compared to the one for pure soft processes ($\propto s^{\Delta_{\text{soft}}}$, $\Delta_{\text{soft}} \simeq 0.1$); the patterns of secondary hadron production in the projectile fragmentation region dominating the EAS development do not differ significantly between soft and semihard inelastic rescatterings.

The latter point deserves an additional discussion. Naively, one may question the importance of relatively high $p_\perp$ jet production for EAS modeling: since such jets are typically produced in the central $y \sim 0$ rapidity region in c.m. frame, having therefore a minor influence on forward hadron production. However, the crucial role here is played by the parton cascades preceding the hardest parton-parton scattering: each previous parton in such a cascade is characterized by a smaller transverse momentum, compared to the subsequent one, $p_{\perp_{i-1}} \ll p_{\perp_i}$, and a higher LC momentum fraction, $x_{i-1} \gg x_i$. Therefore, of highest importance for EAS development are the partons produced in the very beginning of such cascades. In the semihard Pomeron approach, those cascades start already in the nonperturbative region ($p_\perp < Q_0$), hence, at large $x$, thereby having a strong impact on the respective EAS predictions, as discussed in some detail in Ref. [9].

A striking counter-example is the approach of the SIBYLL model [10,11], which ignores the existence of such cascades and takes into consideration the highest $p_\perp$ parton-parton scattering only. Obviously, this is wrong from first principles: those are such parton cascades which produce the above-discussed collinear and infrared enhancements of high $p_\perp$ jet production, being therefore the very reason for the energy-rise of the jet production rate. On the other hand, from the pragmatic point of view, this leads to a serious underestimation of secondary hadron production in the fragmentation regions, giving rise to contradictions with LHC data and to incorrect predictions for EAS characteristics [9].

## 3.2 Microscopic treatment of nonlinear interaction effects

The next crucial step was to consider nonlinear effects related to interactions between the "elementary" parton cascades: treating those as Pomeron-Pomeron interactions and performing all-order resummations of the respective multi-Pomeron graphs [12–14]. This formed the

basis for the development of the QGSJET-II model [15–17], allowing one both to calculate various cross sections for high energy hadronic collisions and to perform MC simulations of inelastic interaction events, generating the (generally complicated) event topologies in strict correspondence with the respective partial cross sections [16].

The corresponding microscopic treatment involved a single additional adjustable parameter, the triple-Pomeron coupling, whose value was constrained based on HERA measurements of diffractive structure functions [15, 16]. On the other hand, it produced a rich phenomenology characterized by numerous nontrivial dynamical effects regarding, e.g. a coherent description of proton structure functions and the energy-dependence of $\sigma_{pp}^{\text{tot}}$ [15], stronger nonlinear effects in proton-nucleus and nucleus-nucleus collisions [16], the energy-dependence of multiparton scattering rates [18] and of the rapidity gap suppression [19].

### 3.3 Higher twist corrections to hard scattering processes

A new theoretical mechanism implemented in the QGSJET-III model [20, 21] concerned a treatment of the so-called power corrections to hard parton-parton scattering processes. In all present MC generators of hadronic collisions, the description of hard processes is based on the leading twist pQCD factorization [22]. In particular, the hardest process in that formalism corresponds to a binary parton-parton scattering involving a single parton from the projectile hadron (nucleus) and a single one from the target. While such a formalism is fully justified for high enough transverse momenta, it is expected to break down at moderately small $p_\perp$, where higher twist corrections become potentially important. Consequently, current MC generators face a problem of an uncontrollable rise of the jet production rate in the small $p_\perp$ limit, which leads to a strong sensitivity of model results to the choice of the above-discussed $Q_0^2$-cutoff separating the treatments of hard and soft processes.

An important class of higher twist corrections corresponding to coherent rescattering of produced $s$-channel partons on "soft" (small LC momentum) gluon pairs has been identified in Refs. [23, 24]. The respective hard scattering processes thus involve arbitrary numbers of partons from the projectile or target hadrons (nuclei). Since the corresponding multi-parton correlators have not been measured experimentally, a model implementation of the approach necessarily implies a phenomenological treatment.

In QGSJET-III, such multi-parton correlators have been interpreted probabilistically: as the so-called generalized multi-parton distributions, which allowed one to develop a dynamical microscopic treatment of the corresponding nonlinear effects, introducing a single additional adjustable parameter $K_{\text{HT}}$ which controls the magnitude of such higher twist corrections [20, 25]. Among the consequences of this new mechanism is a drastic reduction of the model sensitivity to the choice of the $Q_0^2$-cutoff, taming the energy rise of the interaction cross sections and of secondary hadron multiplicity, a stronger damping of low $p_\perp$ jet production in more central collisions, etc. Regarding the parameter $K_{\text{HT}}$, the model results are not too sensitive to its precise value, as illustrated in Fig. 1, where the calculated $\sigma_{pp}^{\text{tot}}$ and $\sigma_{pp}^{\text{el}}$ are plotted both for the default value of $K_{\text{HT}}$ and for $\pm 10\%$ variations of that parameter.

### 3.4 Pion exchange process

An additional technical improvement in the QGSJET-III model concerned a treatment of the pion exchange process in hadronic collisions [21]. The importance of that process for calculations of the EAS muon content $N_\mu$ has been stated in Ref. [17]: in pion-air collisions, the $t$-channel pion exchange enhances forward production of $\rho^0$ mesons, by the expanse of neutral pions, which leads to a $\sim 20\%$ increase of the predicted muon density at ground level. This provides a sufficient motivation to develop a consistent treatment of the mechanism and to cross check the formalism against the data of the LHCf experiment, regarding forward neu-

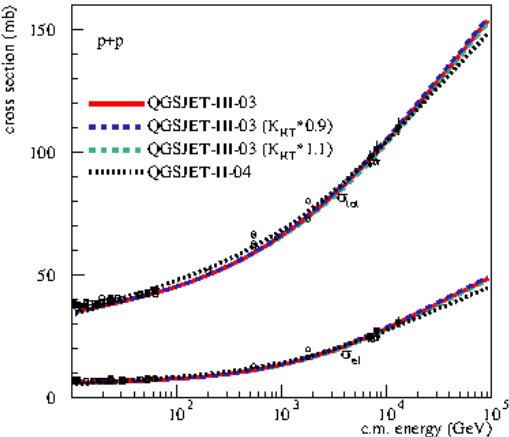

Figure 1: $\sqrt{s}$-dependence of the total and elastic $pp$ cross sections, calculated with the QGSJET-III model, for the default value of $K_{HT}$ and for $\pm 10\%$ variations of that parameter, compared to QGSJET-II-04 results [16,17] and to experimental data.

tron production in $pp$ collisions. The main theoretical challenge here is to predict the energy-dependence of the process, which is governed by the corresponding absorption effects: since those define the probability for not filling the rapidity gap between the forward produced neutron in $pp$ collisions or the forward $\rho^0$ in pion-proton (pion-nucleus) interactions and the other secondary hadrons produced. While the details on the corresponding treatment can be found elsewhere [21], it may be instructive to compare the predicted energy-dependence of forward $\rho^0$ production in pion-nitrogen collisions, between QGSJET-III and other CR interaction models.

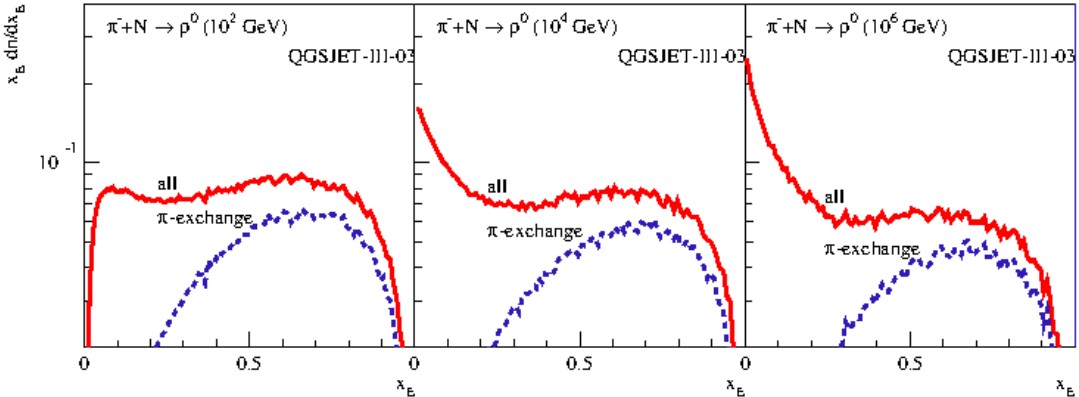

Figure 2: Energy spectra of $\rho^0$ mesons produced in $\pi^- - {}^{14}N$ interactions at $10^2$ (left), $10^4$ (middle), and $10^6$ (right) GeV, calculated with the QGSJET-III model; solid lines - total spectrum, dashed lines - contribution of the pion exchange process.

As one can see in Fig. 2, the pion exchange process in QGSJET-III dominates indeed the forward $\rho^0$ yield and the corresponding contribution slowly decreases with energy, being stronger and stronger damped by the above-discussed absorption effects, which is a direct consequence of the energy-rise of the multiple scattering rate. A similar but stronger damping of the forward $\rho^0$ yield is predicted by the EPOS-LHC model [26], see Fig. 3 (right). On the other hand, in the case of the SIBYLL-2.3 model [11], no pronounced damping with increasing energy is observed for the forward $\rho^0$ yield, as is easy to see in Fig. 3 (left). This indicates that the relevant absorption effects are seriously underestimated, which may lead to an artificial enhancement of the predicted EAS muon content.

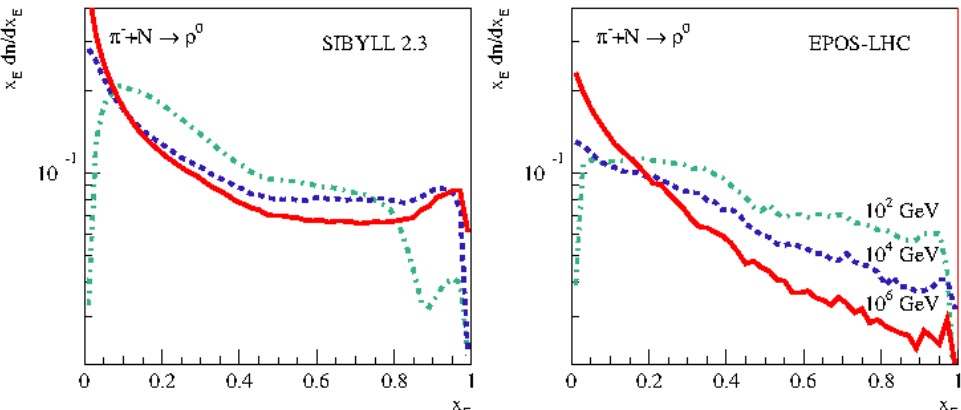

Figure 3: Energy spectra of $\rho^0$ mesons produced in $\pi^- -^{14}N$ interactions at $10^2$ (dashed-dotted), $10^4$ (dashed), and $10^6$ (solid) GeV, calculated with the SIBYLL-2.3 (left) and EPOS-LHC (right) models.

## 4 EAS predictions: QGSJET-III and other models

For basic EAS characteristics, rather small differences have been observed between the predictions of QGSJET-III and of the previous model version, QGSJET-II-04. For example, for the average shower maximum depth $X_{max}$, those amount to some 5 g/cm$^2$, as one can see in Fig. 4. Varying the $K_{HT}$ parameter by ±10% produces only ±2 g/cm$^2$ changes of $X_{max}$. Even smaller

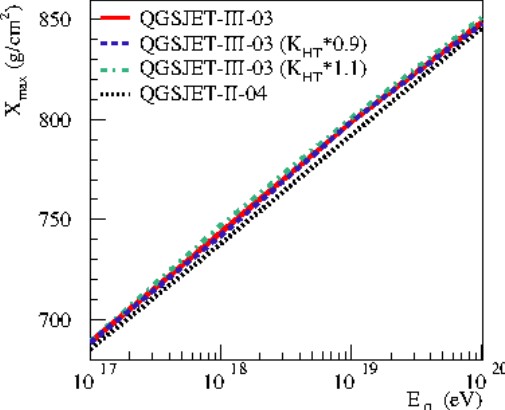

Figure 4: Energy-dependence of the shower maximum depth for proton-induced EAS, calculated with the QGSJET-III model, for the default value of $K_{HT}$ and for ±10% variations of that parameter, compared to the results of QGSJET-II-04.

differences ($\sim$ 1%) between QGSJET-III and QGSJET-II-04 have been observed for the EAS muon content $N_\mu$.

While the reasons for a potential stability of the predicted $N_\mu$ will be discussed elsewhere, let us concentrate here on various model predictions for $X_{max}$. In principle, a robustness of the respective results is expected if: (i) those are dominated by the treatment of proton-air collisions and (ii) such a treatment is sufficiently constrained by LHC data, notably, by measurements of the total and elastic $pp$ cross sections.

Given the higher $X_{max}$ values predicted by the EPOS-LHC and SIBYLL-2.3 models, one may question the validity of both above assumptions. However, a slower EAS development predicted by SIBYLL is a direct consequence of the general theoretical pathology of that model, regarding the treatment of hard processes in hadronic collisions, as discussed in Section 3.1.

More interesting is the case of the EPOS-LHC model: while there seems to be no general problem with the model approach, a number of its technical features appear to be questionable, e.g. an enhanced forward production of (anti)baryons in pion-proton and pion-nucleus collisions. As demonstrated in Ref. [27], the latter has some impact on the predicted $X_{\max}$: shifting the shower maximum depth somewhat deeper in the atmosphere. Since this questions the assumption (i) above, let us have a closer look at the matter.

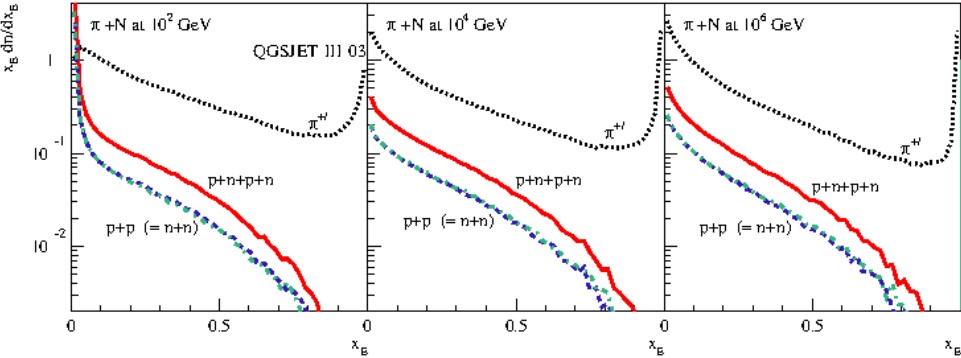

Figure 5: Energy spectra of charged pions and (anti)baryons produced in $\pi^- N$ collisions at $10^2$ (left), $10^4$ (middle), and $10^6$ (right) GeV, calculated with QGSJET-III.

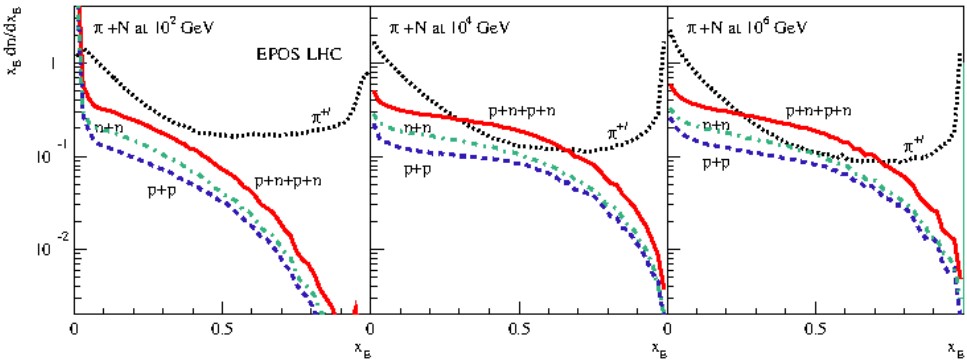

Figure 6: Same as in Fig. 5, calculated with the EPOS-LHC model.

In Figs. 5 and 6, we compare the energy spectra of charged pions and (anti)baryons in pion-nitrogen collisions at $10^2$, $10^4$, and $10^6$ GeV, predicted by QGSJET-III and EPOS-LHC. While in the former model, the forward (anti)baryon yield at all the energies is suppressed by some order of magnitude, compared to the one of pions, a strikingly different behavior is predicted by EPOS: the forward (anti)baryon production steeply rises with energy. Neglecting the very forward ($x_E \to 1$) part of the pion spectra, which is dominated by contributions of diffractive processes, the forward (anti)baryon yield in pion-nucleus collisions, predicted by EPOS-LHC, appears to exceed significantly the one of charged pions, in the very high energy limit. While such a picture can not be excluded by general arguments, it seems to be an artificial one: since there exists no viable theoretical mechanism to produce such an effect.

An additional enhancement of the forward (anti)baryon yield in EPOS-LHC arises from non-respecting the isospin symmetry by that model: as one can see in Fig. 6, its forward $(n + \bar{n})$ spectra exceed substantially the ones of protons plus antiprotons (c.f., Fig. 5 for the respective results of QGSJET-III). Obviously, this is wrong from first principles.

## 5 Relevance to UHECR composition studies

At this point, it is worth to discuss the consequences for experimental studies of the composition of ultra-high energy cosmic rays (UHECRs), arising from a potential robustness of the model predictions for EAS characteristics. It may be instructive to consider the respective results of the Pierre Auger Observatory (PAO), resulting from measurements of the EAS maximum depth $X_{\mathrm{max}}$ [28]. A peculiar feature of the corresponding data is a drastic decrease of $X_{\mathrm{max}}$ fluctuations, $\sigma(X_{\mathrm{max}})$, in the ultra-high energy limit [29]. Consequently, to interpret coherently the experimental measurements of both the average $X_{\mathrm{max}}$ and $\sigma(X_{\mathrm{max}})$, one favors CR interaction models which predict a deeper shower maximum depth or/and smaller fluctuations of $X_{\mathrm{max}}$. In particular, the best overall consistency had been stated for EPOS-LHC, mostly because of the smaller $\sigma(X_{\mathrm{max}})$ predicted by that model for nucleus-induced EAS.

In reality, $\sigma(X_{\mathrm{max}})$ is a theoretically robust quantity, as stated already in Ref. [30]. The small fluctuations of the shower maximum depth, predicted by EPOS-LHC for nucleus-initiated air showers, result from an erroneous treatment of nuclear break up by that model[2] [32,33]. In turn, as discussed in Section 4, the higher $X_{\mathrm{max}}$ values predicted by SIBYLL-2.3 and EPOS-LHC are, at least partly, caused by deficiencies of those models. This rises the question on how the PAO results on UHECR composition would change if the experimental data were reinterpreted using corrected or/and alternative CR interaction models.

## 6 Outlook

We discussed the main theoretical approaches implemented in the QGSJET-III MC generator, comparing also to alternative treatments of the other CR interaction models and revealing a number of serious deficiencies of the latter. As for QGSJET-III, while its predictions are largely driven by the underlying theoretical mechanisms, the model necessarily involves various phenomenological assumptions, which makes its results unwarranted.

Regarding the predictions of QGSJET-III for basic EAS characteristics, those appeared to differ little from the ones of the previous model version, QGSJET-II-04, which may indicate that such predictions are already constrained substantially by available accelerator data, notably, from LHC. In relation to that, we demonstrated that the different EAS predictions of the other CR interaction models can, at least partly, be explained by deficiencies of those models. Therefore, real uncertainties for EAS predictions are very likely smaller than the differences between the results of the present models. A more definite statement requires a thorough quantitative study of such uncertainties, which appears to be an urgent and important task.

## Acknowledgements

**Funding information** This work was supported by Deutsche Forschungsgemeinschaft (project number 465275045).

---

[2]Regarding the importance of the process for the fluctuations of $X_{\mathrm{max}}$, see Ref. [31].

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
