# Peer review of "Cosmic ray interactions in the atmosphere: QGSJET-III and other models"

_SciPost Physics Proceedings, doi:SciPost Phys. Proc. 13, 004 (2023)_

## Round 1 · Referee Report · Anonymous (Referee 1) · 2022-9-12

Strengths

  • clear structure
  • well written
  • very relevant
  • relevance is well explained
  • adequate level of detail for the format of proceedings

Weaknesses

  • sometimes the discussion is too short. It may be clear to experts on hadronic interaction models why certain theoretical concepts are more important than others but to non-experts it just seems like statement vs. statement. A little more detail/background would surely help convincing non-experts .

  • low resolution of figures

Report

In the contribution the author describes new features of their model for hadronic interactions in air showers and compares the predictions and principles of their model with other approaches. Concrete features that are discussed are higher order corrections to QCD factorisation and single pion exchange. In these two processes they identify the differences to other models and highlight their deficiencies. They conclude that their model is the only theoretically consistent one and that using the spread between the different models as a guide for the modelling uncertainty is likely an overestimation and worth further investigation.

The paper is well written, arguments are generally presented most clearly. For a proceeding the level of technical detail is quite adequate. Also the material matches well to what was presented at the conference.

Weak points are: * (minor) The discussion of Fig.2 and Fig.3. Here the same process of rho0 production is shown for three different models at three energies. The point the author makes is the difference in the evolution at large energy fractions, i.e. in the tail of the distribution. But the most striking feature at first glance is the difference in the peak! It would be good if the author could comment here why there is such difference, why it does not matter, etc. A detail: in fig.2 the 1 on all the x-axes is hidden.

  • discussion of fig.5 and fig.6, the case of (anti)baryon production. This is the weakest section. It is basically statement against statement. The author's model predicts a suppression of baryon production with energy while the Epos model predicts a rising production. A little more detail on the principles that lead to the author's prediction may help here (it may be as simple as referring to a previous section). The lack of references in this section suggest that (anti)baryon production is not yet well understood in general. If so that would be worth a statement.

The author also notes the violation of the isospin symmetry in the Epos model, stating it is obviously wrong from first principles. That statement seems to depend somewhat on the first principles one includes. It would be nice to have a little more detail in that claim. For example, these are negative pion-Nitrogen interactions. Couldn't it be that more target neutrons are accelerated? The initial state is not symmetric after all, i.e. pi- + proton might make a neutron but pi- + neutron never makes a proton.

Another thing that could be improved is the quality of the figures. The resolution seems a bit low.

  • validity: high
  • significance: high
  • originality: top
  • clarity: top
  • formatting: excellent
  • grammar: excellent

Author:  Sergey Ostapchenko  on 2022-09-14  [id 2818]

(in reply to Report 1 on 2022-09-12)

I wish to thank the Referee for the review and the positive assessment of my contribution. I address below the remarks of the Referee.

sometimes the discussion is too short. It may be clear to experts on hadronic interaction models why certain theoretical concepts are more important than others but to non-experts it just seems like statement vs. statement. A little more detail/background would surely help convincing non-experts .

I agree with this remark of the Referee. Yet for a conference proceedings, one is forced to choose the statement-like style, given the restricted paper volume. Whenever possible, I supplied the text with references to other papers containing more detailed discussions of the matters.

low resolution of figures

Probably this resulted from the eps-to-pdf conversion required by the paper style.

The discussion of Fig.2 and Fig.3. But the most striking feature at first glance is the difference in the peak! It would be good if the author could comment here why there is such difference, why it does not matter, etc.

Well, the answer is a simple one: the other models don't have any treatment of the pion exchange process and mimic the production of $\rho$-mesons by other processes: hadronization of the proton 'remnant' in SIBYLL model or the usual string fragmentation in EPOS. I preferred to avoid making this statement in my contribution, in order not to be even more critical to the other models, also because it is not that important, compared to the other deficiencies of those models.

My point was to compare the energy dependence of the forward $\rho^0$ yield, whichever way it is predicted by a particular model, because of its relevance for predicting the EAS muon content, and to point on the potential caveat related to disregarding the respective absorptive corrections.

discussion of fig.5 and fig.6, the case of (anti)baryon production. This is the weakest section. It is basically statement against statement. The author's model predicts a suppression of baryon production with energy while the Epos model predicts a rising production. A little more detail on the principles that lead to the author's prediction may help here (it may be as simple as referring to a previous section). The lack of references in this section suggest that (anti)baryon production is not yet well understood in general.

Actually there is no suppression of baryon production in Fig. 5: there is an approximate scaling of the forward yield; the rise of the rapidity plato is not seen in the Figure since plotted is the yield multiplied by $x_E$ in the lab. frame. And the Referee is right: the underlying principles are the same as in Section 3.4 for $\rho$-meson production. Obviously the picture is quite different in Fig. 6. However, I don't see an easy way to explain the underlying picture to the general audience not familiar with the basics of the Regge theory. Yet I find it important to state these differences because of their relevance to the predicted EAS muon content: nobody in the field is aware that the enhancement of the number of muons in the EPOS model results from the assumed energy-dependence of the forward (anti-)baryon yield, illustrated in Fig. 6, rather than from a better model calibration to accelerator data (as claimed by the authors of that model).

The author also notes the violation of the isospin symmetry in the Epos model, stating it is obviously wrong from first principles. That statement seems to depend somewhat on the first principles one includes. It would be nice to have a little more detail in that claim. For example, these are negative pion-Nitrogen interactions. Couldn't it be that more target neutrons are accelerated? The initial state is not symmetric after all, i.e. pi- + proton might make a neutron but pi- + neutron never makes a proton.

The interacting (pion-nitrogen) system is obviously isospin symmetric, apart from the valence quark content of the pion. Regarding the latter, by virtue of the isospin and CPT invariance, the fragmentation of the valence $u$-antiquark into antiproton is identical to the one of the valence $d$-quark into neutron, similarly for $\bar u \rightarrow \bar n$ and $d \rightarrow p$, etc. Thus, while the production spectra of protons and neutrons should be different from each other, this is not the case for $(p + \bar p)$ and (n + \bar n)$.

Anonymous on 2022-09-19  [id 2829]

(in reply to Sergey Ostapchenko on 2022-09-14 [id 2818])

Thank you for the reply. From my side this contribution can be published.

---

## Editorial Decision

published